# Application of Discrete Exterior Calculus Methods for the Path Planning of a Manipulator Performing Thermal Plasma Spraying of Coatings [note 1]

**DOI:** 10.3390/s25030708

**Published:** 2025-01-24

**Authors:** Assel Kussaiyn-Murat, Albina Kadyroldina, Alexander Krasavin, Maral Tolykbayeva, Arailym Orazova, Gaukhar Nazenova, Iurii Krak, Tamás Haidegger, Darya Alontseva

**Affiliations:** 1School of Digital Technologies and Artificial Intelligence, D. Serikbayev East Kazakhstan Technical University, 19 Serikbayev Street, Ust-Kamenogorsk 070010, Kazakhstan; akussaynmurat@edu.ektu.kz (A.K.-M.); akrassavin@ektu.kz (A.K.); tolykbayeva.m@edu.ektu.kz (M.T.); aorazova@edu.ektu.kz (A.O.); nazenova.g@edu.ektu.kz (G.N.); 2Department of Theoretical Cybernetics, Faculty of Computer Science and Cybernetics, Taras Shevchenko National University of Kyiv, 64/13 Volodymyrska str., 01601 Kyiv, Ukraine; iurii.krak@knu.ua; 3University Research and Innovation Center (EKIK), Óbuda University, Bécsi út 96/b, I.–II. em., 1034 Budapest, Hungary; haidegger@irob.uni-obuda.hu

**Keywords:** industrial robot manipulator, 3D scanning, automatic path planning, geodesic distance function, Riemannian manifolds, tangent vector fields

## Abstract

This paper presents a new method of path planning for an industrial robot manipulator that performs thermal plasma spraying of coatings. Path planning and automatic generation of the manipulator motion program are performed using preliminary 3D surface scanning data from a laser triangulation distance sensor installed on the same robot arm. The new path planning algorithm is based on constructing a function of the geodesic distance from the starting curve. A new method for constructing a geodesic distance function on a surface is proposed, based on the application of Discrete Exterior calculus methods, which is characterized by a high computational efficiency. The developed algorithms and their software implementation were experimentally tested with the robotic microplasma spraying of a protective coating on the surface of a jaw crusher plate, which was then successfully operated for crushing mineral-based raw materials.

## 1. Introduction

Nowadays, robot manipulators are widely used for loading, packaging, plasma or laser cutting and melting, as well as for various types of surface treatments: spray painting, plasma spraying coatings, etc. [1,2,3,4,5,6]. The use of a robotic arm to perform coating spraying can increase productivity and repeatability of the process, protect the human operator from the harmful effects of sprayed elements [1,2,3], and in the case of thermal plasma spraying, also protect operators from noise, bright light, and high temperatures [4,5,6].

Concerning the thermal plasma spraying process, the main task performed by the robot arm is to move the plasma source or substrate to expose the substrate surface to the plasma flow with the coating particles in a stable, consistent, and repeatable manner. This means that during plasma spraying, the robot always maintains the same spraying angle (preferably 90°), the given spraying distance (a distance from the plasma source nozzle to the surface), and the fixed distance from which the source or substrate is displaced to bring the next section of the surface into the plasma flow [4,5,6].

From the point of view of the requirements for a uniform distribution of the coating mass on the surface, the plasma spraying process is conducted similarly to the paint spraying process, where the spraying cone moves along the surface. However, there are significant differences between the processes of plasma spraying and paint spraying, as well as in assessing the quality of the final product (e.g., coating), as will be discussed in detail in Section 2 of this article.

Currently, different approaches are used for autonomous offline path generation for robot-assisted spraying applications [7]. One of the first studies in this field was carried out by Suh et al. [8], who developed a theoretical basis for an automatic trajectory planning system for spray-painting robotic arms. A method was proposed for planning the optimal trajectory of the manipulator based on the model of the painting object, providing the optimal generated trajectory (i.e., minimum processing time) and meeting the requirements of uniformity of coating thickness. Trigatti et al. [9] calculated the time parameters of the trajectory of the spray-painting robot, based on the restrictions of the maximum values for speed and acceleration of the end effector imposed by both the characteristics of the manipulator and the technological parameters of the spraying process.

Chen et al. [3,10,11,12] developed a path planning scheme for a spray-painting robot based on the choice of the starting segment of the trajectory as a geodetic line on the sprayed surface. In particular, paper [12] proposed a new path optimization method to improve the efficiency of spraying objects by developing a mathematical model that included the position and direction of the effector and the speed of paint spraying, modeling the workpiece surface, and applying a simulation method surface based on the Flat Patches Adjacency Graph (FPAG). At the same time, the speed of the sprayer and the width of the overlapped area were calculated to set the trajectory of the robot, and the spraying trajectory was planned for each area. The dispersion of the paint thickness of a discrete point and the ideal paint thickness were taken as the objective functions, and the trajectory in each section was optimized. It has been proven that the proposed trajectory optimization of a 3D robot paint sprayer can fully satisfy the requirement of spray thickness uniformity.

The basic idea of the method proposed by Atkar et al. [13] was to divide the surface into areas with simple topology patches, but these patches should not have the shape of a ring, figure eight, etc. In addition, these patches must be geodetic convex in a trajectory planned for each patch separately. Thus, the trajectories consisted of segments of equidistant curves, so that, on a plane, they would represent straight segments. These curves were obtained by parallel transfer along the surface of the initial segment of the seed curve. The uniform coating was achieved by choosing the geodetic distances between the trajectory fragments and the time parameters for passing these sections. The method proposed in paper [13] was called an offset curve planner approach in a review [7] by Weber et al.

Zhou Yu. et al. [14] emphasized that the shape of the workpiece can strongly influence the quality of the sprayed coating. Fu et al. [15] proposed a surface segmentation method based on a genetic algorithm in which a complex surface was divided into several surfaces of small curvatures to simplify the trajectory planning operations for robotic spraying of complex surfaces. Chen et al. [12] used a segmentation method based on dividing the triangular mesh of the 3D solid surface, setting the maximum normal vector threshold, and connecting the triangle surface into a smaller flat area according to the triangle connection algorithm. As a result, each patch is approximately planar, and at least one side of each patch is part of the 3D solid ridge. Zhou B. et al. [16] developed an algorithm for surface slice-projection processing to automatically generate the manipulator working tool trajectory, and the simulation results demonstrated the algorithm’s effectiveness. Zeng et al. [17] developed a model of the rate of the coating’s growth using the technology of spraying with a different angle of inclination; experiments have shown that the model had good accuracy.

Xia et al. [18] aimed to provide plasma spraying of uniformly thick coatings on surfaces with complex geometry. The use of a programmable robot made it possible to plan the plasma spraying trajectory or change processing parameters, such as the spraying angle and distance, scanning speed and step, etc. As a result, the authors of paper [18] developed a robotic plasma spraying system with intelligent adaptive adjustment of the trajectory of the robotic arm and control of the manipulator in real time. Thus, robotic plasma spraying has proven to be a feasible and highly effective solution, ensuring high process precision and repeatability of coating applications.

Cooper et al. [19] proposed the “mesh following technique” for generating robot tool paths based on tessellated surfaces with irregular edges and holes. McGovern and Xiao [20] proposed a technique based on tessellated surfaces to create a full coverage trajectory using a mesh projected onto a surface; however, it is limited to the basic shapes and the individual parts should be considered as a single patch.

Recently, several new approaches have been proposed in autonomous tool-path planning and paint-spraying simulations, in which either the point cloud is included as a geometric representation of the model [1,2] or to obtain the model the geometry is generated using any existing CAD software or involves the use of sensors such as depth cameras, light detection and ranging (Li-DAR), and coordinate measurement [7]. Thus, Nieto Bastida and Lin [1] developed an algorithm for autonomously generating robot trajectories based on input 3D point cloud data that surrounds an object with a predefined spherical mesh that organizes geometric attributes into a structured data set. The point cloud model is visualized and serves as the basis for automatic path planning (robot code generation), while a 3D sensor is used to localize the position of the workpiece in front of the robot and adjust the trajectory of the manipulator.

Nowadays, the literature describes methods for generating a manipulator trajectory for spray painting optimized for various types of 3D models. For example, for the common STL 3D model format, Wu et al. developed a boundary-fitting approach [21]. The raw 3D scanning data are a point cloud. The methodology of manipulator path generation based on point cloud segmentation is presented in the article by Hua et al. [22]. A very promising approach using a machine vision system for online manipulator path generation when scanning 3D objects was presented by Mac et al. [23]. It should be noted that 3D surface metrology, sensing, point cloud data generation/analysis, 3D surface generation, and path planning and trajectory generation are effectively similar for painting [8,9,10,11,12,14,15,16,21], fiber placement [24,25], laser surfacing, polishing, and plasma spraying [18]; therefore, special attention is paid to the consideration of these approaches, in particular the approach used for trajectory generation for robotic fiber placement [25].

Currently, the use of robotic manipulators in industry is usually limited to large-scale production, since each transition to a new type of product requires complex calibration procedures to comply with the model embedded in the robot during its manufacture [1,2,26,27,28]. Therefore, the task of automatically generating a program code for a robotic arm from a CAD model is the focus of researchers and developers of robotic systems. The practical implementation of such a task could make it possible to efficiently process the surfaces of small-scale and piece products of complex shapes using a robot manipulator.

The main idea of this research is to develop an intelligent robotic system for the plasma processing of industrial products, which makes it possible to implement the automated pass planning of a robot arm. A distinctive feature of the proposed intelligent robotic system is a preliminary 3D scanning of the surface of the workpiece with subsequent automatic generation of the robot manipulator program code, taking into account the data of the 3D scan of the object (Figure 1). Pre-scanning is carried out by distance sensors mounted on the same robotic arm that sprays plasma on the object.

Thus, the trajectory plan must be generated for the thermal spraying process without any initial knowledge of the product shape or orientation. In the literature, the research closest to the main ideas of this study are the different algorithms developed by Nieto Bastida and Lin [1], Atkar et al. [13], and Chen et al. [3,10,11,12] for the operation of an intelligent robot in an environment with uncertainty. In this case, the trajectory is formed by the robot control system based on information about the current state of the external environment, that is, according to the 3D model of the processed surface reconstructed by the robot, which is a point cloud (coordinates of the object surface).

The implementation of such an intelligent robotic system would make it possible to process pieces and large-sized products, the geometric parameters of which are determined with low accuracy or products with deviations from a given shape. At the same time, the methods of robotic 3D scanning and scanning data processing previously developed in papers [29,30,31] are inexpensive compared to machine vision methods, since they use cheap distance sensors. It must be highlighted that this method is quite accurate due to the use of an industrial robot with precise positioning of the working tool. The idea of using such an intelligent robotic system for thermal plasma spraying of coatings and the sequence of operations shown in Figure 2, as well as brief information on the implementation of new control algorithms for performing robotic plasma spraying of a protective coating on the surface of an industrial product of complex shape, were presented at the 11th IEEE International Conference on Intelligent Data Acquisition and Advanced Computing Systems: Technology and Applications (IDAACS-2021), Cracow, Poland, 22–25 September 2021, and published in paper [29].

This study focused on developing an automatic path planning for a robot manipulator performing microplasma spraying of coatings. Two significant aspects distinguish the method proposed in this paper from other path planning methods:Some methods for forming the trajectory of a manipulator for applying coatings involve changing the speed of the working tool (the spray gun). We assume that the robot’s working tool moves along the surface of the work area at a constant speed.Some other methods for forming a manipulator trajectory are united by a common concept, which is sometimes called the “seed curve offsetting method” [7]. This method involves two primary steps: first, identifying the optimal starting curve (also known as the seed curve), and second, generating subsequent strokes by offsetting them across the entire surface of the workpiece [7].

The significant difference of our proposed method consists of using the geodesic distance field calculation procedure. We propose a method in which the value of the function $\phi_{\gamma }$ is calculated at each mesh node, and then individual segments of the trajectory trace are constructed as isolines of the scalar field on the meshes. We do not resort to the procedure of “shifting” the starting curve, which, in our opinion, provides certain advantages.

There are several known methods for constructing the geodesic distance function from the starting curve, but their applications are still difficult [7]. In this paper, we propose an original new method based on the use of Discrete Exterior calculus methods, which is more computationally efficient.

## 2. Experiments

### 2.1. Features of Thermal Plasma Spraying and Requirements for the Trajectory of the Plasma Source to Ensure Uniform Coating Thickness

Thermal plasma spraying (TPS) of coatings is widely used in mechanical engineering and the automotive industry, the aerospace industry [32], and, more recently, in medicine [4,5,6,33]. TPS makes it possible to form coatings with controlled thickness for various purposes from a wide range of materials (metals, alloys, and ceramics) by carefully selecting and maintaining appropriate spraying parameters. The coating thickness can range from 50 µm to 1000 µm, depending on the spraying parameters and the number of passes of the plasma jet with the coating material along the surface of the workpiece (that is, the substrate). Atmospheric TPS is a method of applying coatings to the surface of parts using plasma obtained in a gas environment. This method involves introducing small particles of materials, such as metals, ceramics, or polymers, into a plasma jet generated by ionizing a gas, usually argon, through an electric arc. The main feature of thermal plasma spraying is the high temperature in the plasma jet (>20,000 °K). Powders of refractory metals and alloys or ceramics melt in a plasma jet and deform when hitting the surface and solidify, forming a coating that can improve various surface characteristics such as hardness, wear resistance, corrosion resistance, and biocompatibility. TPS coating always has a lamellar structure and contains a certain number of internal pores. The number and size of these pores also depend on the spraying parameters and can thus be controlled [5,6]. The key requirements for TPS ensure the production of a uniform thickness coating with good adhesion to the substrate. The requirements are the perpendicular incidence of the plasma jet on the surface of the substrate, accurate adherence to the specified distance between the plasmatron nozzle and the surface of the substrate, and uniform movement of the plasma source or substrate along a trajectory that ensures overlap spraying paths by a given amount. The spraying distance significantly affects the microstructure and adhesion of the coating. If the distance from the plasma nozzle to the surface being treated is too large, the molten particles may cool down before reaching the substrate and may not adhere well to it.

Also, a controlled spraying parameter is the speed of movement of the plasma source along the surface of the sprayed product since the thickness of the coating depends on this. The thickness of the coating is made greater by lowering the selected moving speed, while other spraying parameters remain unchanged, such as electric current strength, plasma gas flow rate, etc. The plasma source is usually moved using a robotic manipulator. From the point of view of ensuring the safety of the TPS process, the advantages of using a robot are obvious since the process applies high temperatures and is accompanied by noise, bright light, and air pollution with particles of the sprayed material and substrate, as well as the need to move a heavy plasma source (a plasmatron). The same considerations concern plasma welding and cutting processes [28].

Currently, one of the promising TPS methods is microplasma spraying (MPS). MPS allows the application of coatings from both powder and wire materials to substrates made from a variety of materials. Due to the small diameter of the spray spot, ranging from 3 mm to 5 mm, the loss of sprayed material during MPS is significantly less than during conventional TPS. Due to the low power of the microplasmatron (up to 2 kV), the thermal effect of the MPS process on the substrate is minimal, which makes it possible to obtain coatings on thin-walled and small-sized parts without their deformation and overheating and special cooling of the substrate. In addition, the mass of the microplasmatron is significantly (up to 10 times) less than that of a conventional plasmatron, and the noise level of the MPS process is also considerably lower.

By selecting the MPS parameters, it is possible to obtain a coating with the desired thickness, with satisfactory adhesion to the substrate, and even to a certain extent control the porosity of the coating and its surface roughness [5,6]. Here, we should indicate such MPS parameters that are not controlled through a robotic manipulator but are set by the coating developers and are maintained constant through the settings of the MPS installation. These parameters are the following: electric current (I in amperes [A]), plasma gas flow rate (Q, standard liter per minute [slpm]), spraying distance (H, millimeters [mm]), powder feed rate (V_pow_, grams per minute [g min^−1^]), and wire feed rate (V_w_, meters per minute [m min^−1^]). It is important to mention that such a parameter as the speed of movement of the plasma jet along the surface of the substrate V (mms^−1^) is provided by the control of the manipulator, in contrast to the material feed rate into the plasma jet (V_w_ or V_pw_) provided by the settings of the MPS installation. The main TPS parameters, the maintenance of which is the goal of robotization of the TPS process, are the spraying distance H (mm), the speed of movement of the plasmatron along the surface of the substrate V (mms^−1^), and the spraying angle. The uniformity of the coating thickness is ensured by planning the trajectory of the plasma source. Regarding requirements for uniform distribution of coating on the surface, the TPS process has similar features to the paint spraying process, in which the spraying cone moves over the surface of the part. As proven by Chen et al. [10,11], the distribution profile of paint on the substrate surface can be described by a complex shape curve, namely the envelope curve of Gaussian profiles formed at a fixed distance from each other. Trigatti et al. [9] examined the surface profile of spray paint, carried out 3D modeling of the coating profile obtained by a moving source, and concluded that the material’s distribution in a moving spray cone is also close to Gaussian. Thus, to ensure uniform coating thickness, the spraying trajectory should be selected to ensure a fixed distance between the vertices of the spraying cones, which correspond to the maxima of the Gaussian distribution. In this case, the spraying path formed by moving the base of the spraying cone along the surface partially overlaps. Thus, when spraying a uniform coating, the optimal trajectory of the spraying cone is a U-shaped curve, and the return passage of the spraying source should be offset by a fixed distance, which is called the spray step.

Let us explain the analogy with the paint spraying cone using the example of MPS coatings and also clarify the selection of the spraying step. As was experimentally proven in [34,35], when using a stationary plasmatron that sprays a coating onto a stationary substrate (see Figure 2), the distribution of the material in the “metallization figure” (see Figure 3) obeys a Gaussian distribution.

**Figure 2 sensors-25-00708-f002:**
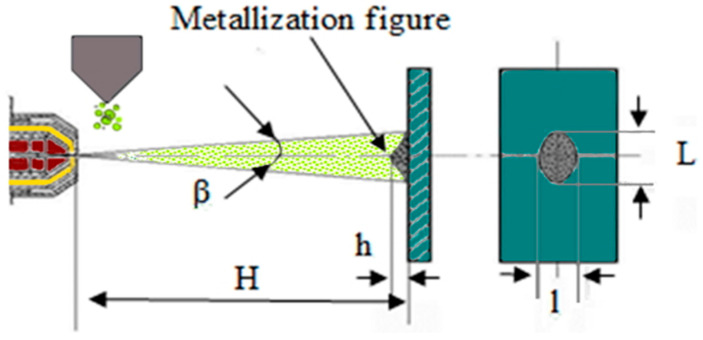
Diagram illustrating the measurement of the metallization figure, where **H** represents the spraying distance, **β** denotes the opening angle of the plasma jet, **h** represents the height of the sprayed material, **L** refers to the vertical (large) axis, and **l** indicates the horizontal (small) axis of the spray spot [34].

Metallization figures were produced by microplasma spraying of TiO_2_ powder [34] or Zr wire [35] (Figure 3) onto steel substrates, with the microplasmatron held in a fixed position for 10 s. The dimensions of the metallization figures were evaluated by measuring the lengths of the vertical (major) axis (L) and the horizontal (minor) axis (l) of the spray spot, along with the maximum height (h) of the deposited material (Figure 2).

The metallization figures were captured using an Olympus 460 digital camera (Olympus, Tokyo, Japan) from angles perpendicular to their axes. The images were then processed to determine the coordinates of the metallization figure profile. Using these coordinates, the profiles of the metallization figures were obtained with the use of the Mathcad Prime 3.0 (PTC, Montreal, QC, Canada) software, and functions describing the profiles were determined for the large and small axes of the spray spot. This allowed for the calculation of the areas of the metallization figures. Analysis of these profiles revealed that the shape of the figures during MPS can be accurately described by a Gaussian function or normal distribution, as shown in Equation (1) [36]:(1)$$y=y_{0}e^{-kx^{2}}$$
where $y_{0}$ represents the maximum height of the metallization figure; k is a numerical coefficient; and x is the width of the metallization figure. The correlation coefficients for the calculated and actual curves ranged from 0.9849 to 0.9992, with the value of k in Equation (1) varying between 0.12 and 0.97 [34,35].

The width of the spray path (the diameter of the base of the spraying cone) depends on the spraying distance. When spraying with a moving plasma source, the coating is formed as a result of the superposition of single rollers of metallization figures in the process of linear movement of the spraying spot at a speed V relative to the sprayed surface (see Figure 4). We believe that the distribution of the material in metallization figures is still described by a Gaussian distribution, as was shown in paper [9], while the height of the metallization figure determines the maximum thickness of the coating. In contrast, a coating with a minimum thickness is formed in the overlap area of the bases of the metallization figures (Figure 4).

As shown in Figure 4, a uniform coating, which is a coating for which the minimum difference between its largest (δ_max_) and smallest (δ_min_) thicknesses is ensured, is formed by observing a certain spraying step τ. In the case of TPS, the best uniformity of coating thickness distribution is achieved under Condition (2) [37]:τ ≤ 0.35 L (2)
where L is the width of the spraying path, i.e., the diameter of the metallization figure cone base (see Figure 2).

Thus, during TPS of the coating, the plasmatron must move along a U-shaped path, while the return passage of the plasma jet must be offset by a fixed distance equal to the spraying step τ, which is approximately equal to one-third of the spraying path width. To set the spraying step τ, it is necessary to experimentally measure the width of the coating spraying path on the substrate at the given TPS parameters and then calculate the step τ.

During the TPS of a coating, it is necessary to maintain a specified spraying distance. Thus, the robot must move the plasmatron at a given distance from the surface along a trajectory that follows the shape of the surface of the part, then turn around, shift horizontally by the spraying step, and move in the opposite direction, again along the surface with precise adherence to the spraying distance.

### 2.2. Research Equipment and Materials

The implementation of the automatic path planning based on the 3D model of the scanned object and the motion planning was carried out for the Kawasaki RS010L (Kawasaki Robotics, Akashi, Japan) robot manipulator controlled by the E40F-A001 programmable controller with the use of AS (Advanced Superior) software [38]. The characteristics of the robotic manipulator (Kawasaki RS-010LA) are as follows:Positioning accuracy: 0.06 mmMaximum linear speed: 13,100 mm/sEngagement zone: 1925 mmWorking load capacity: 10 kg

The motion of the robot manipulator is completely determined by the program in the AS (Advanced Superior) language, which is executed by the robot’s control system. The high-level robot control language AS can be viewed as an end-effector motion description language. The AS program describes the motion of the center of the working tool as motion through the number of path positions and also describes the spatial orientation of the end effectors. High-level control software solves reverse kinematic tasks for given end-effector motion descriptions, as well as other trajectory planning tasks. All low-level control tasks (such as force and position control) are performed by the manipulator controller.

The preliminary scanning of substrates was carried out by an ODSL 8/D4-400-S12 (Leuze electronic, Owen, Germany) laser triangulation distance sensor installed on the manipulator. In general, the choice of a triangulation laser sensor for this study was due to the above-mentioned considerations, such as the ability to carry out non-contact measurements without using a video camera with sufficiently high accuracy both in a static position and in dynamic mode, as well as the low cost of the sensor compared to machine vision systems. According to the supplier [39], the ODSL 8/D4-400-S12 sensor features a measurement range of 20 to 400 mm with a resolution of at least 0.1 mm. Its radiation source operates within the 560–580 nm wavelength range (visible red light), and the light spot size is at least 1 × 6 mm at a distance of 400 mm. The absolute measurement error does not exceed 1% for distances up to 200 mm and not worse than 2% for distances up to 400 mm. The repeatability (as a percentage of the measured value) is at least 0.25% at distances of up to 200 mm and minimally 1% at distances up to 400 mm, and the nonlinearity is no more than 0.5%. The sensor offers a maximum measurement frequency of at least 200 Hz, with a measurement time of not less than 5 ms and a response time not exceeding 20 ms. It supports digital interfaces such as RS485 or RS 232; the RS232 operates with a supply voltage in the range of 18–30 V and consumes a maximum current consumption of not more than 100 mA.

Schematically, the connection between the components of a three-dimensional scanning system using a non-contact distance sensor installed on a robotic manipulator is shown in Figure 5.

After completing the 3D scanning procedure, the MPS-004 microplasmatron (E.O. Paton Institute of Electric Welding, Kyiv, Ukraine) weighing 1.2 kg was mounted on the industrial robotic arm and moved to perform microplasma spraying of metal powders onto metal substrates. The speed of linear movement of the microplasmatron along the substrate (V) was chosen to be 40 mms^−1^. This speed was chosen experimentally to ensure plasma spraying of the coating with a uniform thickness. Experiments have shown that this speed of linear traveling of the microplasmatron does not lead to disturbances in the plasma jet flow due to air resistance and, therefore, ensures the stability of the spraying process with different parameters.

Austenitic high-manganese Hadfield steel (grade 110G13L) was used as the substrate material. The elemental composition of steel 110G13L is given in Table 1. The steel was manufactured by the VostokMashZavod JSC (Ust-Kamenogorsk, Kazakhstan). The chemical composition of the steel specimens was provided by the manufacturer and tested by energy-dispersive X-ray analysis (EDX) using a JSM-6390LV scanning electron microscope (JEOL, Tokyo, Japan) with an energy-dispersive analysis attachment (Inca Energy, Oxford Instruments, Abingdon, UK).

The powder of Cr-based composite alloy AN-35 was used for MPS. Coatings were applied to the surface of laboratory steel specimens with dimensions of 20 × 30 × 10 mm^3^ and to the surface of the crushing plate by robotic MPS of AN-35 composite alloy powder, which was used for surfacing and spraying wear-resistant coatings on parts, machines, and equipment operating under conditions of abrasive wear, corrosion, erosion, elevated temperatures, and aggressive environments. The elemental composition of AN-35 powder is given in Table 2.

The movable plate of a jaw crusher is a large-sized part weighing 295 kg made of Hadfield steel with the following dimensions: 900 × 820 × 100 mm^3^. The plate has a ribbed surface with a distance of 100 ± 11 mm between the centers of the ribbed projections, and the height of the projections is 30 ± 1 mm.

MPS of coatings on steel substrates was carried out with the parameters indicated in Table 3. Before MPS, the surface of laboratory specimens and the crushing plate were sandblasted.

With the selected MPS parameters (Table 3), the width of the spraying path was 5.5 mm, and the spraying step τ, in accordance with Formula (2), was chosen to be 1.8 mm. The coating thickness varied from 200 µm to 400 µm due to changes in the number of passes of the plasma jet.

The choice of MPS parameters listed in Table 3 was determined by the requirements for porosity, hardness, adhesion, and wear resistance of the coating. Accordingly, studies of the microstructure and tests of the mechanical properties of coatings were carried out on laboratory specimens; however, a description of all the details of this materials science study is not relevant here and therefore not given. It is significant that as a result of this materials science study, such MPS parameters were selected as the microplasmatron linear movement speed **V**, spraying distance **H**, and spraying step **τ**, which should be kept constant during the MPS process.

## 3. Results and Discussion

### 3.1. D Scanning and Surface Reconstruction Procedures

During the scanning procedure, the trajectory of the manipulator working tool (the distance sensor) lies in the horizontal plane above the scanned surface. The trajectory consists of connected U-shaped segments (Figure 6).

Work segments of the trajectory are parallel straight-line segments that cover the scanned area. When passing the working segments of the trajectory, the distance sensor data are read. The robot manipulator controller allows to read the current spatial position of the working tool on request, and the laser triangulation distance sensor also allows to perform measurements on request.

In the process of scanning, a couple of requests are sequentially executed from the computer:A request to perform measurements on the distance sensor.A request to transfer the coordinates of the working tool to the controller of the robotic arm.

The time interval between these two requests is so small that it is assumed that the distance was measured exactly at the location of the sensor, which is determined by the coordinates received as a reply to the second request.

The result of the scanning procedure is a point cloud. The scanning results data are highly noisy, so converting a point cloud into a triangular mesh is a rather complex procedure, which is described in detail in previous papers [29,30,31].

### 3.2. Manipulator Path Planning

The input to the path planning procedure is a model of the processed surface in the form of a triangular mesh obtained at the scanning stage and the output is a program in AS language.

When the working tool of a robotic manipulator moves along the surface being processed, the intersection point of the working tool axis with the surface will move along a curve on the surface, which we will call the trace of the working tool on the surface. Of course, for a given surface, the manipulator path, i.e., a given curve in space along which the center of the working tool moves with the spatial orientation of the working tool specified at each point of this path (the direction of the axis of the working tool), uniquely determines the trace of the path on the surface. We assume that the orientation of the working tool is directed normally to the surface at the point of intersection of the axis of the working tool and the surface and that there is a certain rule for calculating the distance on the axis of the working tool from the point of intersection of the axis with the surface to the center of the working tool (in the simplest case, this distance is the given fixed distance from the nozzle to the surface being processed). Then, the trace of the working tool uniquely determines the trajectory of the working tool. Thus, the problem of path planning is reduced to the problem of determining a set of curves (in the general case) on the surface corresponding to the trace of the working tool on the surface being processed.

Thus, the path planning procedure for the manipulator can be divided into three stages:Constructing a trace of the manipulator path on the surface.Constructing a set of points lying on the path of the manipulator’s working tool (thus, the trajectory of the working tool is specified by a broken line).Transformation of the geometrically specified path of the manipulator’s working tool into the manipulator program in AS language.

The main research of this study is focused on the first stage since the second and third stages are quite simple. The description of some subtle issues related to the representation of the trajectory of a working tool in the form of a sequence of geometric primitives (circular arcs and straight segments) was discussed earlier in paper [29]. Of course, the general approach to manipulator trajectory planning, which involves constructing the manipulator path on a surface, is quite common for robotic spraying applications in painting, varnishing, etc., as mentioned in the Introduction section. However, the problem of constructing the manipulator path on the surface is closely related to the problems arising in the tasks of robotic fiber placement, more precisely, there are a number of trajectory planning tasks for resin transfer molding (RTM), automated tape placement (ATP), and robotic fiber placement (RFP) [25]. The interrelations between these trajectory planning methods and the proposed method will be discussed in more detail in the next section.

#### 3.2.1. Offset Curve Planner Approach and Geodesic Distance Function

The trajectory planning method we propose can be classified as an offset curve planner approach [7]. This approach was introduced by Atkar et al. in paper [13]. There are two main steps in this method: first, to determine an optimal starting curve (also called the seed curve), and second, to generate the next strokes by offsetting them along the entire surface of the workpiece. The seed curve should preferably be a geodesic, although this is not compulsory. A detailed description of the method and issues related to the choice of the starting curve can be found in paper [13] and in the comprehensive review in [7]. Then, the following offset curves are determined by calculating an orthogonal geodesic to the seed curve and generating a new stroke at a uniform geodesic distance, which ensures that the strokes are equidistant at all points. Similarly, the Surface Curve Offset (SCO) algorithm for RFP presented by Shirinzadeh et al. in [25] is based on an approach that can be summarized as follows: “For a given RFP path, the next path must be offset along the surface by a distance of one tow-width in the perpendicular direction from the given path”. The distinctive feature of the SCO algorithm is that the offset of a given point of the curve along the surface (in a given direction) is determined by the intersection of the so-called control plane with the surface, and the Newton–Raphson root-finding method is used to solve the surface–plane intersection equation (strictly speaking, the initial path is also formulated from the surface–plane intersection). Thus, the SCO algorithm proposed by Shirinzade et al. in article [25] is applicable to the problem of constructing trajectory traces and may achieve better results in terms of accuracy than previously proposed curve shift methods [13].

We propose a method in which the offset curves are constructed as isolines of the geodesic distance function $\phi_{\gamma }$. For the given curve $\phi_{\gamma }$ on the surface S, we will denote a scalar function $\phi_{\gamma }:S\to R$, which we refer to as the geodesic distance function, which is defined as follows: for any p∈S, $\phi_{\gamma }\left(p\right)$ is the geodesic distance from p to γ.

Thus, the proposed method consists of three stages:Determination of the seed curve γ on the surface.Numerical calculation of the geodesic distance function $\phi_{\gamma }$ defined on the surface.Construction of isolines $\phi_{\gamma }=kd$ of the geodesic distance function $\phi_{\gamma }$, where k is an integer index, and the given value of the parameter d represents the distance between consequent strokes.

It is easy to see that the offset curves in the method proposed by Atkar [13] are essentially isolines of the geodesic distance function from the seed curve. Accordingly, the method of constructing offset curves can be considered as an implicit numerical method for constructing isolines of the distance function. Instead, we propose an explicit method in which the distance function values are first calculated at each grid node, and then the isolines of the calculated distance function are constructed using standard methods. This approach is a significant improvement over the original offset curve planner approach in terms of the accuracy of constructing offset curves. The main drawback of the original approach [13] is the accumulation of errors during the consistent application of the offset curve generation procedure. Another important requirement for the procedure of constructing offset lines is high computational performance. Most of the known methods for calculating the geodesic distance function are characterized by relatively low performance. If these methods are applied to construct offset curves, the computational performance of such an algorithm will be lower than that of the original implementation of the offset curve planner approach [7,13]. This relatively low efficiency of the known methods was the main motivation for the development of a new method for calculating the geodetic distance function presented in this paper.

Since the distinctive feature of the proposed trajectory planning method is the use of the geodesic distance function, we will focus on the problem of calculating this function from a given curve (seed curve). This particular task belongs to a broad class of problems involving the computation of geodesic paths and distances on a polyhedral surface. It is a well-developed area of applied mathematics that continues to attract researchers’ attention (particularly due to its practical significance). The problems of computing geodesic distances are divided into two classes: computing geodesic distances from a single point (Single Source Geodesic Distance, SSGD) or from multiple points, which can be geometric objects such as curves or flat figures on the surface (Multiple Source Geodesic Distance, MSGD). Many methods for computing geodesic distances reduce the problem of solving partial differential equations on a smooth manifold. When working with surfaces represented by meshes, the equation is discretized and the finite element method (FEM) or mesh method is used to solve it. These methods are applied to both SSGD and MSGD problems, and often the same method can be applied to both classes of problems by differently setting the boundary conditions for one differential equation. Methods that reduce the problem of solving differential equations (PDE-based methods) are divided into two groups: Wavefront-based methods and Diffusion-based methods [40].

The most straightforward approach to computing the geodesic distance function is solving the Eikonal Equation (3) for given boundary conditions:(3)|∇ϕ|=1

Most methods in the first group implement this approach. The Eikonal Equation (3) can serve as a formal definition of the geodesic distance function and has a clear physical meaning (if we consider the special case of the Eikonal Equation (3) as an equation specifically for the distance function). However, it is a nonlinear hyperbolic equation, which is extremely difficult to solve directly in many cases.

Methods in the second group operate on parabolic-type equations, offering significant computational advantages and enabling the creation of efficient software implementations. The problem with these methods is that they calculate, at best, only an approximation of the geodesic distance function. A breakthrough in the field of geodesic distance function computation methods based on numerical solutions of differential equations was the emergence of the heat flow method [41]. With sufficiently high computational efficiency, this method allows (subject to certain requirements for the quality of mesh triangulation) the calculation of the distance function with reasonable accuracy.

When attempting to apply geodesic distance function computation methods based on solving differential equations to our task, a specific problem arises. This lies in defining the boundary conditions on the boundary of the domain. In general, at least for methods like heat flow [41], this problem is solvable but requires the use of certain artificial techniques. In this paper, we propose a new method for calculating the geodesic distance function on a surface, which is specifically designed to compute the distance from a curve defined in the plane, free from such problems. The method is based on the energy minimization procedure of a vector field proposed in paper [42]. The application of Discrete Exterior calculus methods, suggested by the authors of paper [42] for the computational implementation of their method, offers, in our opinion, a significant advantage of our proposed method over traditional PDE-based distance function computation methods in terms of implementation efficiency. Before proceeding to the algorithm description, we will discuss the main mathematical ideas and constructions underlying the proposed method.

#### 3.2.2. Isolines of Geodesic Distance Function as Integral Lines of Vector Fields

Most methods for finding the geodesic distance function involve solving partial differential equations on smooth manifolds. We propose a method for calculating the geodesic distance function based on computing a vector field on the surface whose integral lines coincide with the isolines of the function $\phi_{\gamma }.$ This interpretation of the problem has a clear motivation. We can interpret the geometric procedure of constructing the isoline $\phi_{\gamma }$ = *l* as a physical process. Let us imagine we have two small pucks, *A* and *B*, connected by an inextensible string of length *l*. We move puck *A* along the starting curve ***γ***, while moving puck *B* across the surface such that two conditions are met: Firstly, the string must always be taught and lie on the surface. If we associate the pucks with two points *A* and *B* on the surface, we can say that the string lies on the segment of the geodesic line *A**B*, and the geodesic distance between points *A* and *B* is maintained during the motion. Secondly, the direction of the geodesic line *A**B* at point *A* is orthogonal to the tangent of ***γ*** at point *A*. The trajectory of point *B* on the surface will be a curve, which we will denote as $p_{l}$. In general, the curve $p_{l}$ may have self-intersections and even intersect with the curve ***γ***. Also, even for smooth ***γ***, curve $p_{l}$ may not be smooth everywhere. However, we can assume that for a segment of the curve ***γ***, a parameter value *lmax* > 0 is defined, such that if *l* ∈ [0, *lmax*], then $p_{l}$ will represent a smooth curve without self-intersections, on which a smooth one-dimensional tangent vector field is defined. By varying the value of the parameter *l*, we obtain a smooth tangent vector field ***U***, which is defined in a region of the surface adjacent to the starting curve ***γ***.

Figure 7 schematically illustrates the kinematic procedure for constructing the isolines of the vector field function $\phi_{\gamma }$ and the corresponding tangent vector field ***U***.

For the simplest case—when the starting curve γ is given in the plane—we can easily obtain an analytical expression for the corresponding field U. Let us assume that puck A moves along the curve γ with a constant unit speed, i.e., $v_{A}=\dot{\gamma }$. The system of two pucks connected by a string can be considered a rigid body. If we denote Ω as the angular velocity vector of this rigid body and denote $r_{AB}=r_{B}-r_{A}$, where $r_{B}$ and $r_{A}$ are the radius vectors of points A and B, respectively, then for any time moment (t) Equation (4) holds:(4)$$v_{B}\left(t\right)-v_{A}\left(t\right)=\Omega \left(t\right)\times r_{AB}$$

It is easy to see that the angular rate of rotation of the unit vector $v_{A}$ Is equal to the angular speed $\left|Ω\right|$ of the system of two pucks, which are considered as a rigid body. Thus, for any t, Equation (5) holds true:(5)$$\left|\Omega \left(t\right)\right|=\left|\frac{d^{2}}{ds^{2}}\gamma \left(s\left(t\right)\right)\right|$$

It is well known that the magnitude and direction of the vector $\frac{d^{2}}{ds^{2}}\gamma$ have a simple geometric interpretation, namely (6):(6)$$\left|\frac{d^{2}}{ds^{2}}\gamma \left(s\right)\right|=k\left(s\right)=\frac{1}{R\left(s\right)}$$
where $k\left(s\right)$ is the curvature of γ at the point $\gamma \left(s\right)$, and *R*(*s*) is the radius of the circle tangent to the curve γ at the point $\gamma \left(s\right)$. The vector $\frac{d^{2}}{ds^{2}}\gamma \left(s\right)$ is orthogonal to the tangent vector $\dot{\gamma }\left(s\right)$. It can be said that the vector $\frac{d^{2}}{ds^{2}}\gamma \left(s\right)$ points towards the center of the tangent circle O. The point *O* is also the instantaneous center of rotation of the virtual “rigid body” (mechanical system consisting of two pucks connected by a string). Let us define a scalar function $\chi \left(s\right)$ on the curve γ such that $\left|\chi \left(s\right)\right|=k\left(s\right)$, and the sign of $\chi \left(s\right)$ is defined by Equation (7):(7)$$sign\left(\chi \left(s\right)\;\right)=\left\{\begin{array}{c} 1\;,\;if\;\left(\frac{d^{2}}{ds^{2}}\gamma \left(s\right),r_{AB}\right)\leq 0\\ -1\;\;\;\;\;\;\;\;\;\;\;\;\;\;\;\;\;\;\;\;\;\;\;otherwise \end{array}\right.$$

Taking into account Equations (5) and (6) and the definition of “signed curvature” $\chi \left(s\right)$ given above, one can easily derive Equation (8) from Equation (4):(8)$$v_{B}\left(t\right)=\left(1+l\chi \left(s\left(t\right)\right)\right)\dot{\gamma }\left(s\left(t\right)\right)$$

The corresponding vector field U for the coordinates s,l are given by Formula (9):(9)$$U\left(s,l\right)=\left(1+l\chi \left(s\right)\right)\dot{\gamma }\left(s\right)$$

#### 3.2.3. Isometric Flows of Tangent Vector Fields

In fact, the kinematic procedure described above for constructing the tangent field U on the surface defines a function $U\left(\gamma ,v\left(s\right)\right)$. The concept of a flow of a vector field will play a key role in the analysis of the vector fields $U\left(\gamma ,v\left(s\right)\right)$. Formally, the flow of a tangent vector field on a smooth manifold is defined as a one-parameter group of mappings of the manifold onto itself.

**Definition 1.** *Let* U *be a tangent vector field defined on a smooth manifold M. The vector field defines a parameterized set of mapping* $\psi^{t}:M\to M$ *. For a given value of the parameter* t *, the action of the mapping* $\psi^{t}$ *on a point* p∈M *is defined as* $\psi^{t}\left(p\right)=\gamma_{p}\left(t\right)$ *, where* $\gamma_{p}\left(t\right)$ *is the solution of the differential Equation (10) with the initial condition* $\gamma_{p}\left(0\right)=p:$(10)$$\frac{d}{dt}\gamma_{p}=U\left(\gamma_{p}\left(t\right)\right)$$

In courses on the theory of ODEs (ordinary differential equations) it is proven that, under certain conditions imposed on the smoothness of the field U (which we will assume to be fulfilled), the solution to Equation (10) exists and is unique. The curves γ are called integral curves of the vector field U. Just as linear transformations of the plane, the mappings $\psi^{t}\left(p\right)$ form a group with respect to the composition operation. In particular, it is easy to show that $\psi^{t}\left(\psi^{s}\left(p\right)\right)=\psi^{t+s}\left(p\right)$ and $\psi^{0}\left(p\right)=p$. The one-parameter group of mappings $\psi^{t}:M\to M$ is called the flow of the tangent vector field U on the manifold M.

This definition will be required later for constructing analytical procedures. Still, to show the main ideas on which the proposed method is based, it is more convenient to use physical interpretations of the concept of the flow of a vector field. First, if we use the hydrodynamic interpretation of the concept of a flow of a vector field, we will immediately see the following fact. The flow of the vector field $U\left(\gamma ,v\left(s\right)\right)$ has a property invariant to the choice of the starting curve γ and the distribution $v\left(s\right)$; if we place an elastic filament in the flow, so that at any point the line of the filament is orthogonal to the streamlines of the field, the length of the filament will not change with time as it moves in the flow. Moreover, in some cases, the flow of the tangent vector field $U\left(\gamma ,v\left(s\right)\right)$ can be isometric. Let us imagine that we place a small piece of a thin film made of elastic material on a surface where a tangent vector field U at the point ***p*** of the surface over which point *q* is located at that moment in time. In general, the piece of film will move in such a flow, rotating and deforming as it moves. But if the vector field ***U*** on the surface is such that no matter where on the surface the piece of film begins its movement it will move along the surface without deformation, the flow of such a field will be isometric. Let us give a simple example. Let the surface be a segment of a sphere of radius R, centered at point O. As the starting curve γ we choose the arc of the circle formed by the intersection of the surface α, passing through the point O with the sphere. Let us call p the line passing through the center of the sphere O and perpendicular to the surface α. If we construct the level line of the function of the geodesic distance $\phi_{\gamma }$, moving the puck A along the arc γ at a unit speed, then this is equivalent to the fact that we would rotate the sphere around the axis p with an angular velocity $\omega =\frac{1}{R}$, while keeping the pucks A and B so that they remain motionless, and the filament stretched. It is clear that the corresponding vector field U on the segment of the sphere coincides with the field of velocities of the points of the sphere arising when the sphere rotates around the axis p with an angular velocity $\omega =\frac{1}{R}$. Obviously, the flow of such a vector field will be isometric. Tangent vector fields whose flows are isometric are called Killing vector fields. The given example is a classic example of a Killing vector field on the surface of a sphere.

The converse statement can be readily proven. Let V be a Killing vector field on a smooth two-dimensional surface. Denote γ as an arbitrary integral line of this field. Then, any integral line of the field V will be an isoline of the geodesic distance function $\phi_{\gamma }$. For a special case, characterized by the fulfillment of two conditions formulated below, this statement allows us to analytically formulate the problem for a tangential vector field such that if U is its solution, then the integral lines of the field U will be isolines of the geodesic distance function $\phi_{\gamma }$.


On the surface we are working with, Killing vector fields exist. It should be noted that Killing vector fields do not exist on all surfaces.As the initial curve γ we choose an integral line of some Killing field on the surface.


Subsequently, we will extend this analytical problem to the general case of a smooth two-dimensional surface by introducing the concept of an approximate Killing vector field. For this, we will need some definitions from the field of Riemannian geometry.

#### 3.2.4. Two-Dimensional Surfaces Embedded in Three-Dimensional Space as Riemannian Manifolds and Analytical Definition of Killing Fields

Henceforth, we will consider a smooth two-dimensional surface embedded in three-dimensional Euclidean space $R^{3}$ as a smooth Riemannian manifold $\left(M,g\right)$ with the metric induced by the Euclidean metric in $R^{3}$. This last statement implies the following definition of the metric on the manifold M. Let p∈M be an arbitrary point on the surface and $\left(\varphi ,U\right)$ be a chart of the manifold M, such that p∈U, and then we choose some local coordinate system $x_{1},$ $x_{2}$ in the region $\varphi \left(U\right)$.

If we choose a Cartesian coordinate system in the three-dimensional space into which our surface is embedded, with origin at an arbitrary point O, then we define a local parameterization of the surface in the neighborhood of the point p of the form $r\left(x_{1},x_{2}\right)$, where $r\left(x_{1},x_{2}\right)\in R^{3}$ is the radius vector of the point $\varphi^{-1}\left(x_{1},x_{2}\right)$ in this coordinate system. In this case, the vectors s $u,v\in R^{3}$, defined by the equalities $u={\left(\frac{\partial r}{\partial x_{1}}\right)}_{p}$ and $v={\left(\frac{\partial r}{\partial x_{2}}\right)}_{p}$, are invariant under shifts of the point O and form a basis in the surface tangent to the surface at the point p. The Gram matrix of the basis v,u will be the matrix of the metric tensor $g_{p}$ at the point p in the local coordinates $x_{1},$ $x_{2}$ (11):(11)$$g_{p}=\left(\begin{array}{cc} \left(u,u\right) & \left(u,v\right)\\ \left(v,u\right) & \left(v,v\right) \end{array}\right)$$

Note: If we denote as E,F,G the values at the point p of the coefficients of the first fundamental form on this surface, calculated in local coordinates $x_{1},$ $x_{2}$, then the equalities $E=\left(u,u\right)$, $F=\left(u,v\right)$,$\;G=\left(v,v\right)$ will hold, and we can say that the induced metric g is given by the first fundamental form on the surface. At any point p∈M the metric g defines a scalar product $\left(v,u\right)=g_{p}\left(v,u\right)$ $v,u\in T_{p}\left(M\right)$ in the tangent space $T_{p}\left(M\right)$. When a metric is given on a smooth manifold, the concept of an isometric flow of a vector field and, along with it the concept of a Killing vector field, can be defined as follows:

**Definition 2.** *A tangent vector field* U *on a smooth Riemannian manifold M will be a Killing vector field if and only if for any point* p∈M *and any two tangent vectors* $X,Y\in T_{p}\left(M\right)$ *the following Relation (12) holds:*(12)$$g_{p}\left(X,Y\right)=g_{\psi \left(p\right)}\left(d\psi_{p}\left(X\right),d\psi_{p}\left(Y\right)\right),$$*where* $\psi =\psi^{t}$ *is the flow of the vector field* U *for* $t\in \left(-\varepsilon ,\varepsilon \right)$ *for some small* ε>0 *, and d is the differential of the mapping* ψ:M→M*.*

From Definition 2, it is easy to obtain Equation (13), which is often considered as the definition of a Killing vector field U on a smooth Riemannian manifold:(13)$$L_{U}g=0,$$
where L is the Lie derivative.

#### 3.2.5. The Laplace–de Rham Operator and the Relation Between Killing and Harmonic Fields on Two-Dimensional Surfaces

Using the intuitive physical interpretation of the concept of an isometric flow of a tangent vector field, it is easy to formulate two necessary conditions that a Killing vector field V on a two-dimensional surface embedded in three-dimensional space must satisfy:The geodesic curvature of any integral line of the field V must be constant.On any curve ξ, which is an integral line of V, the condition $\forall p\in \xi \left|V\left(p\right)\right|=const$ must hold.

Therefore, the vector field $U\left(\gamma ,v\left(s\right)\right)$, constructed using the above-described kinematic procedure, can only be a Killing vector field if it satisfies Condition (14) and the geodesic curvature of γ is constant:(14)$$\forall p\in \gamma \;\;U\left(p\right)=\dot{\gamma }$$

Let us now try to reverse our perspective on Condition (14). We will consider it as a condition imposed on the vector field in an analytically formulated problem. Note that Condition (14) uniquely defines a Killing vector field on the surface (if there exists a Killing vector field on the surface such that γ is its integral line). This fact is intuitively obvious if we use the physical interpretation of the concept of an isometric flow of a vector field. If we consider the surface as a smooth Riemannian manifold, then a stronger statement holds: a Killing field is determined uniquely by a vector at some point and its gradient (i.e., all covariant derivatives of the field at the point). A number of important insights for constructing an analytical procedure for constructing vector fields associated with the geodesic distance function are provided by considering vector fields $U\left(\gamma ,v\left(s\right)\right)$ on the surface. Earlier we saw that on the surface, the field ***U*** is given by the distribution of curvature $\chi \left(s\right)$ along the initial curve γ. Consider the simple but important case where the curvature of γ is constant. On the surface, there are two types of curves with constant curvature: straight lines and arcs of circles. The two types of initial curves—straight lines and arcs of circles—correspond to two types of vector fields on the surface. Straight lines correspond to constant vector fields $C_{k}$, which are analytically defined by Formula (15):(15)$$\forall p\in R^{2}\;\;C_{k}\left(p\right)=k$$
where k is the unit vector (the direction vector of the straight line γ). If γ is an arc of a circle with a center at point O, then the field U will coincide with the velocity field of points of the rotating disk $R_{\omega }$. Let us choose the center of the Cartesian coordinate system at point O and imagine that the axis of the disk rotating with angular velocity $\omega =\left|\chi \right|=k$, where k is the constant curvature of the arc of the circle and γ is projected onto this point. In such a coordinate system, the components $R_{x}$, $R_{y}$ of the field $R_{\omega }=R_{x}i+R_{y}j$ are given by Formula (16):(16)$$R_{x}\left(x,y\right)=\omega y\;R_{y}\left(x,y\right)=-\omega x$$

It is evident that if the flow of the field U on the surface is isometric, then the field must be divergence-free, i.e., $\forall p\in R^{2}\;\nabla \cdot U\left(p\right)=0$. As for the curl of the field, as it is easy to check $\left|\nabla \times R_{\omega }\right|=2\omega$, and for vector fields U on the surface whose flows are isometric, we can formulate the requirement of the constant curl of the field $\forall p\in R^{2}\;$∀*p*
$\nabla \times U\left(p\right)=\Omega_{c}$, where $\Omega_{c}$ is a constant vector perpendicular to the plane under consideration. It follows that any Killing vector field on the plane will belong to the kernel of the Hodge operator (sometimes called the Laplace–de Rham operator), which is defined based on $R^{2}$ by Equation (17):(17)⧠=∇∇·+∇×∇×

As can be easily seen, if X is a harmonic field on the surface, i.e., Condition (18) hold for X, then X belongs to the kernel of the Laplace–de Rham operator, i.e., Equation (18) follows from the fulfillment of Condition (19):(18)∇·X=0  ∇×X=0(19)⧠X=0

Let us note that the fields $C_{k}$ are both harmonic and Killing. The fields $R_{\omega }$, obviously, are not harmonic, but as will be shown below, they are closely related to harmonic fields. Let us remind you that an arbitrary harmonic vector field is uniquely defined by its restriction on the arbitrary segment of its flow line. Let ξ be an integral curve of a harmonic vector field V on the surface, and let s be the natural parameter of the curve ξ, so $\forall p\in \xi V\left(p\right)=v\left(s\right)\dot{\xi }$. If the function $v\left(s\right)$ is given on an arbitrary interval $\left[s_{1},s_{2}\right]$, then this function can be used to reconstruct the vector field V over the entire plane. In other words, the harmonic vector field is fully determined by its restriction to a segment of the streamline. Accordingly, if the starting curve γ is a segment of the integral line of some harmonic vector field X, then the vector field X is uniquely determined by Condition (14). Let the condition $\forall p\in \gamma \;X\left(p\right)=\dot{\gamma }\left(p\right)$ be given on the arc γ of a circle of radius R centered at the point O for the harmonic vector field X. In a Cartesian coordinate system, centered at the point O, the components $X_{x}\left(x,y\right)$ and $X_{y}\left(x,y\right)$ of the harmonic vector field $X=X_{x}i+X_{y}j$ are determined by Equation (20):(20)$$X_{x}\left(x,y\right)=\frac{y}{\left(x^{2}+y^{2}\right)}\;\;\;\;X_{y}\left(x,y\right)=-\frac{x}{\left(x^{2}+y^{2}\right)}$$

The integral lines of this field will be arcs of circles centered at the point O. Thus, if U is a Killing vector field and X is a harmonic vector field on the surface, given by the condition $\forall p\in \gamma \;U\left(p\right)=X\left(p\right)=\dot{\gamma }\left(p\right)$, where γ is a curve of constant curvature, then the integral lines of the fields X and U coincide. Let us extend these results to the general case of curved two-dimensional surfaces. First, let us refer to the definition of the Laplace–de Rham operator acting on smooth tangent vector fields. We will consider the most general case of an arbitrary Riemannian manifold $\left(M,g\right)$. We will denote the linear space of the smooth tangent vector fields on M as $X\left(M\right)$. Then, the Laplace–de Rham operator $◻:X\left(M\right)\to X\left(M\right)$ is defined by Formula (21) [43]:(21)$$◻=S+\Delta_{r},$$
where $\Delta_{r}$ is the rough Laplacian operator and S is the Ricchi operator, defined by Equation (22), where Ric is the Ricci tensor:(22)$$Ric\left(X,Y\right)=g\left(SX,Y\right)\;\;X,Y\in X\left(M\right)$$

It is known that if U is a Killing vector field on a Riemannian smooth manifold $\left(M,g\right)$ or if U is a harmonic vector field on M, then ◻U=0 [44].

Let us pose the following Problem 1: find a tangent vector field X on a two-dimensional Riemannian manifold $\left(M,g\right)$ such that ◻X=0 and $\forall p\in \gamma \;X\left(p\right)=\dot{\gamma }\left(p\right)$, where γ is a smooth curve on M.

As follows from the above, either this problem will have no solution, or the set of solutions ${\bar{X}}_{\gamma }$ will consist of vector fields of form (23):(23)$$X_{\alpha }=\alpha U_{\gamma }+\left(1-\alpha \right)Y_{\gamma }\;\;X_{\alpha }\in {\bar{X}}_{\gamma }$$
where $U_{\gamma }\in T\left(M\right)$ is a Killing vector field defined by the condition $\forall p\in \gamma \;U_{\gamma }\left(p\right)=\dot{\gamma }\left(p\right)$, $Y_{\gamma }\in T\left(M\right)$ is a harmonic vector field defined by the condition $\forall p\in \gamma \;Y_{\gamma }\left(p\right)=\dot{\gamma }\left(p\right)$, and α∈R is a parameter. The integral lines of the fields $U_{\gamma }$ and $Y_{\gamma }$ coincide and represent the level curves of the geodesic distance function $\phi_{\gamma }$. Due to this, for any value of the parameter α the integral lines of the vector field $X_{\alpha }$ will also be level lines of the function $\phi_{\gamma }$, and we can consider the set ${\bar{X}}_{\gamma }$ as a set of vector fields whose integral lines are level lines of the function $\phi_{\gamma }$. Let us consider now the case of an arbitrary starting curve γ on the surface, without assuming that the curvature of γ is constant. We will assume that γ is parameterized by the natural parameter s. Assume that on some interval $I=\left[s_{1},s_{2}\right]$, and thus Condition (24) holds:(24)$$\forall s\in I\;\;\chi \left(s\right)=c+\Delta \chi \left(s\right)$$
where c∈R is a given constant and $\forall s\in I\;\left|\Delta \chi \left(s\right)\right|<\varepsilon$, where ε>0 is a small parameter. Then, in the surface region $\left[s_{1},s_{2}\right]\times \left[0,l_{max}\right]$, the field $U\left(\gamma \right)$ will be approximately Killing. Analytically, this means that for any ∀s∈I, the magnitude of $\left|◻U\left(s\right)\right|$ is small. It should be noted that the known problem is approximating a curve with two ends on the plane of a piecewise smooth curve consisting of a finite number of smooth segments, each of which is either a straight-line segment or a circular arc. This issue is often formulated as the problem of the decomposition of a planar curve into arcs and line segments. Problems of such decomposition, in a certain sense, optimal decomposition, arise in digital image processing, approximation of curves by splines of certain types, and in robotics and several methods of such decomposition are currently known [45,46,47]. If the seed curve on the plane allows approximation by a piecewise smooth curve, which is a sequence of straight-line segments and circular arc segments, then the vector field *U* corresponding to this starting curve will be the “approximate Killing vector field” in the above sense. Assuming that we can draw an analogy between the above-considered case of a curve on a plane and the general case of a two-dimensional surface embedded in a three-dimensional Euclidean space, we consider the segmentation of a two-dimensional surface, in which the surface is divided into “geometrically homogeneous regions”, and that within each such region of homogeneity the values of Gaussian and average curvature of the surface are approximately constant. Segmentation of this kind has been described previously in articles [45,46,47]. If a geodesic line is chosen as the starting curve on the surface, then within each region of homogeneity, the vector field *U* determined by the starting curve will be the approximate Killing vector field, in the sense of the relative smallness of the value |◻*U*|. As the results of numerical experiments with both scanning data and synthetic data show, in the vast majority of cases when applying the segmentation method described above, most of the surface area is covered by such regions of homogeneity. However, the question of which conditions should be imposed on the surface to guarantee that any geodesic seed curve corresponding tangent vector field can be considered as an “approximate Killing vector field”, remains open.

#### 3.2.6. The Energy Functional of the Vector Field

As it is known, constructions of classical differential analysis of vector fields, such as the gradient of a scalar function, divergence, and curl of a vector field, have their analogs and generalizations in the calculus of smooth differential forms on smooth manifolds, which are sometimes called exterior calculus. In many cases, operating with dual forms of the first order (covector fields) to vector fields is much easier than directly with vector fields. In our case, the application of the language of k-forms will have, as will be seen from the following, great advantages. Below, we present basic information about the relationship between vector fields and covector fields and introduce the notation to be used in the further exposition.

The metric defines a canonical isomorphism on the manifold (which is usually called the musical isomorphism) between the tangent bundle $T\left(M\right)$ and the cotangent bundle $T^{*}\left(M\right)$, which is given by two mappings (25):(25)$$b:T\left(M\right)\to T^{*}\left(M\right),\;\#:T^{*}\left(M\right)\to T\left(M\right)$$

For any point p∈M and any vector $v\in T_{p}\left(M\right)$, the covector $v^{b}\in T_{p}^{*}\left(M\right)$ is defined so that Condition (26) is satisfied:(26)$$\forall u\in T_{p}\left(M\right)\;\;v^{b}\left(u\right)=g_{p}\left(v,u\right)$$

Correspondingly, at an arbitrary point p∈M the inverse transformation $v^{\#}\in T_{p}\left(M\right)$ for an arbitrary covector $v\in T_{p}^{*}\left(M\right)$ is defined so that Condition (27) is satisfied:(27)$$\forall u\in T_{p}\left(M\right)\;\;v\left(u\right)=g\left(v^{\#},u\right)$$

We will denote the spaces of smooth k-forms on the manifold M as $\Omega_{k}$. In this case, as is known, $\Omega_{0}$ is the space of smooth scalar functions on M (the space of smooth mappings f:M→R) and $\Omega_{1}$ is the space of smooth covector fields on M (i.e., each element $T^{*}\left(M\right)$ will be an element of $\Omega_{1}$ and vice versa; each element of $\Omega_{1}$ will be an element of $T^{*}\left(M\right)$).

The Laplace–de Rham operator $◻:T\left(M\right)\to T\left(M\right)$, which we considered above, is the operator dual to the Hodge–Laplace operator, acting in the space $\Omega_{1}$ dual to $T\left(M\right)$ (28):(28)$$\forall U\in T\left(M\right)\;\;◻U={\left(\Delta U^{b}\right)}^{\#}$$

The Hodge–Laplace operator $\Delta :\Omega_{k}\to \Omega_{k}$ is defined in (29):(29)$$\forall U\in T\left(M\right)\;\;◻U={\left(\Delta U^{b}\right)}^{\#}$$
where $d:\Omega_{k}\to \Omega_{k+1}$ is the exterior derivative operator and $\delta :\Omega_{k+1}\to \Omega_{k}$ is the so-called codifferential. In those cases where the operator δ acts on k − 1 forms of the space $\Omega_{1}$, Formula (30) holds:(30)$$\forall X\in T\left(M\right)\;divX=\delta X^{b}$$

Thus, the codifferential can be considered as a generalization of the divergence operator. The exterior derivative operator d can be considered as a generalization of the curl of a vector field since Formula (31) holds:(31)$$\forall X\in T\left(M\right)\;rot\;X={\left(*dX^{b}\right)}^{\#}$$
where ∗ is the Hodge star operator. A harmonic form is a form $\omega \in \Omega_{k}$, satisfying Condition (32):(32)δω=0     dω=0

Of course, with such a definition, the harmonic form $\omega \in \Omega_{1}$ will be the form dual to the harmonic vector field. The space of harmonic forms on a manifold gives the kernel of the Hodge–Laplace operator (for any harmonic form Δω=0).

The authors of paper [41] defined the energy functional of the vector field as a mapping E:Ω_1→R, the action of which on the *k* − 1—form (covector field) ω is determined by Formula (33):(33)$$E\left(\omega \right)=\langle \omega ,\Delta \omega \rangle$$

It is evident that if $\omega \in \Omega_{1}$ is a harmonic form, then $E\left(\omega \right)=0$. The authors of paper [41] arrived at the construction of functional (33) by considering the bilinear form on covector fields E, defined as $E\left(\omega ,\xi \right)=\langle d\omega ,d\xi \rangle +\langle \delta \omega ,\delta \xi \rangle$. If $U\in T\left(M\right)$ is an arbitrary tangent vector field on M, then the magnitude $E\left(U^{b},U^{b}\right)$ can be considered as a quantitative criterion for the dissimilarity of the vector field U from the harmonic field, interpreting the term $\langle \delta U^{b},\delta U^{b}\rangle$ is the integral divergence of the field U, and the term $\langle dU^{b},dU^{b}\rangle$ as the integral curl of the field U. The task of minimizing the functional $E\left(U\right)$ under given constraints on the vector field U can be interpreted as the problem of finding a vector field that, while satisfying the imposed conditions, is as close as possible to the harmonic field. Further, we assume that the vector field U is subject to only one constraint of the form $\forall p\in \gamma \;U\left(p\right)=\dot{\gamma }\left(p\right)$, where γ is a geodesic line on the surface. If, for a given surface and a given initial curve γ, there exists a set of solutions to task 1 ${\bar{X}}_{\gamma }$, then by virtue of (28) $\forall X_{\alpha }\in {\bar{X}}_{\gamma }\;E\left(X_{\alpha }^{b}\right)=0$. These facts and the considerations presented in the previous section lead to Proposition 1:

**Proposition 1.** *If the covector field* $U\in \Omega_{1}$ *minimizes the functional* $E:\Omega_{1}\to R$ *, defined by Formula (31) under the condition* $\forall p\in \gamma \;U\left(p\right)={\dot{\gamma }}^{b}$ *, then the integral lines of the vector field* $U^{\#}\in T\left(M\right)$ *will coincide with the isolines of the function* $\phi_{\gamma }$*.*

Since we do not have proof of Proposition 1 for the general case of arbitrary surfaces in three-dimensional space, Proposition 1 in the form stated above should be considered a hypothesis. The proofs of this proposition for some specific cases that we have obtained are not rigorous, but the intuitively convincing considerations that led to the formulation of this hypothesis and the results of numerical experiments make the assumption of the validity of this hypothesis highly probable in our opinion. Until a proof (or disproof) of the validity of this hypothesis is obtained, we consider the numerical method for calculating the geodesic distance function based on this hypothesis as a heuristic method that provides good practical results and is convenient to use.

#### 3.2.7. Remarks on the Numerical Implementation of the Method

The proposed method for constructing the geodesic distance function is based on a computational point of view on the numerical methods for minimizing the energy functional of a vector field proposed by Fisher et al. [42]. The authors of article [42] proposed a new approach to the problem of designing tangent vector fields with user-defined constraints based on the application of Discrete Exterior calculus methods. Previously proposed approaches to the vector field design problem by various authors were based on the general idea of interpolating vectors specified in individual mesh nodes representing the surface. All these approaches rely on explicit coordinate frames and vectors represented through coefficients in these frames, which are either 2D or 3D. The parallel transport of tangent vectors between these coordinate frames generally makes vector field optimization a nonlinear problem. In contrast, in article [42], the vector field design problem was formulated as a linear problem by using an intrinsic, coordinate-free approach based on discrete differential forms and the associated Discrete Exterior calculus methods. The use of Discrete Exterior calculus techniques—in which covector fields are represented by scalar quantities on the edges of the mesh and operators acting on differential forms by matrices—allows us to construct highly efficient algorithms in the software implementation, where it is convenient to use advanced libraries of numerical methods of linear algebra. A detailed presentation of numerical methods for minimizing the energy functional of a vector field is given in article [42]; here, in the following presentation, the focus is on aspects specific to the problem of constructing the geodesic distance function.

The algorithm for constructing the distance function consists of a sequence of three procedures:Constructing a geodesic line on the surface, which we refer to as the initial curve γ.Calculating the vector field U, which minimizes the energy functional of vector Field (33) under Condition (14).Reconstructing the distance function from the vector field U, whose integral lines approximately represent the isolines of the function $\phi_{\gamma }$.

We work with a discrete representation of the surface using a triangular mesh. The triangular mesh is programmatically represented as a linear list of vertices (each vertex corresponds to a data structure representing a tuple of three elements consisting of the coordinates of the vertex in space) and a so-called triangle list. Each triangle corresponds to one of the faces of the polyhedron that describes the surface. Each triangle is represented as a tuple of three elements consisting of the indices of the triangle’s vertices in the vertex list. Naturally, the vertex and triangle list indirectly define the set of edges of the polyhedron representing the surface. We will denote the vertices as $v_{i}$, where i is the index of the vertex in the list. The edge connecting the vertices $v_{i}$ and $v_{j}$ will be denoted as $e_{ij}$, where the order of the indices indicates the orientation of the edge. The triangles (mesh faces) will be denoted as $t_{ijk}$, where we will assume that the order of the vertex indices determines the orientation of the triangle. The covector $\omega_{1}$, dual to a given tangent vector field u on the surface, is represented by its values on the edges $c_{ik}=\int_{v_{i}}^{v_{k}}udl$; the values $c_{ik}$ theoretically represent the integral of u along the edge $e_{ik}$. Step *B*) is implemented as a program module that takes as input a triangular mesh and a list L of tuples $\left(i,j,value\right)$, specifying the predetermined values $c_{ij}$. The list L provides a discrete representation of boundary Condition (1). In the described program implementation of the method for constructing the geodesic distance function on a surface, the user selects two points, A and B, on the boundary of the region M. Then, a discrete representation of the geodesic line connecting points A and B is constructed on the mesh. The representation of the geodesic line as a path through the faces of the surface is then converted into the list L and fed into the program module that implements step B). The code for this module implements numerical methods for minimizing the energy functional of the vector field, as detailed in article [42]. To reconstruct the tangent vector field on the surface from the calculated discrete covector field, the Whitney finite elements method is applied. The module provides an interface that allows obtaining the coordinates of the tangent vector to the surface at a given point, based on the barycentric coordinates of the point on a specified face of the mesh. The first stage of the procedure for reconstructing the geodesic distance function $\phi_{\gamma }$ step C) is the construction of the tangent vector field $g\phi^{*}$, which is an approximation of the gradient field $\nabla \phi_{\gamma }$. The construction of the vector field $g\phi^{*}$ is carried out in three steps:In the center of each face $t_{ijk}$ (triangle of the mesh), the vector $u_{ijk}$ is calculated.Each vector $u_{ijk}$ is rotated by an angle of $\frac{\pi }{2}$ in the surface of the face. The choice of turning direction (clockwise or counterclockwise) is indifferent. We will interpret this step as the computation of the discrete field $p_{ijk}=J\left(u_{ijk}\right)$, where J is a linear rotation operator by $\frac{\pi }{2}$.For each face, a vector is calculated as follows: ${g\phi }_{ijk}^{*}=\frac{p_{ijk}}{\left|p_{ijk}\right|}$.

Thus, we have a standard problem in numerical methods: reconstructing a scalar function defined on a surface from the gradient field of the function. We use a standard approach. The problem is formulated as finding the minimum of the functional D with respect to $\phi^{*}$, which is an approximation of the function $\phi_{\gamma }$ that we are seeking:(34)$$D\left(\phi^{*}\right)=\int_{M}{\left|\nabla \phi^{*}-g\phi^{*}\right|}^{2}$$

As is known, the Euler–Lagrange equation for this extremal problem has the following form (35) [48]:(35)$$D\Delta \phi^{*}=div\;g\phi^{*}$$

The previously computed discrete tangential vector ${g\phi }_{ijk}^{*}$ defines a Dirichlet boundary condition on the boundary ∂M. However, as practice has shown, it is better to solve the problem with mixed boundary conditions separately for the two regions into which the initial curve γ divides the original region M. In this case, for both regions, the initial curve γ will be part of the boundary where the Cauchy boundary condition $\forall p\in \gamma \phi^{*}\left(p\right)=0$ is satisfied.

To test the software implementation of the described method for constructing the geodesic distance function from a curve, we used both synthetic data (triangular meshes calculated for analytically defined surfaces) and data obtained from scanning. It should be noted that most of the available 3D mesh libraries (such as MeshLib) do not contain functions that calculate the geodesic distance from a given curve on the surface. Typically, these libraries provide functions that calculate the geodesic distance between two given points on the surface or, at best, numerically calculate the geodesic distance function from a given grid vertex. Thus, using independently developed libraries to quantitatively assess the accuracy of calculating the geodesic distance function using the algorithm described above is difficult. For such a comparative assessment, we used our own software implementation of the geodesic heat method. We extensively used the library of computational geometry methods to create a software implementation of the geodesic heat method [49]. As shown by the results of numerical experiments, the relative error in calculating the geodesic distance from a point on the surface to a given curve using the algorithm described above in most cases does not exceed 0.7% and is practically independent of the position of the point relative to the curve. Considering that we compare two approximate methods, our algorithm and a geodesic heat method, we believe that the accuracy of the proposed method is more than sufficient for trajectory planning tasks. Note that from a practical point of view, for complex surfaces, a good choice of the seed curve has a much greater impact on the final result of the surface processing procedure than the accuracy of calculating the geodesic distance function.

Software implementation of the third stage (the translation of the sequence of geometric primitive into the sequence of AS language commands) is straightforward, and will not be discussed here; it was previously described in articles [29,30,31].

### 3.3. Experimental Application of the Developed Control Algorithms

The developed method of automatic manipulator program generation was tested and the robotic MPS of a wear-resistant Cr-based coating was performed on the worn parts of the jaw’s crushing plate for crushing mineral raw materials. A robotic scanning of the crushing plate was carried out, and then a robotic MPS of a protective coating on the worn-out sections of the plate was performed. A visual representation of the sequence of the process of robotic MPS of the powder coating on a crushing plate, including preliminary 3D scanning, is given in Figure 8.

Specifically, for scanning the crushing plate (Figure 8a), a scanning step of 30 mm was used, the speed of movement of the end effector was 100 mms^−1^, the distance between the scanning plane and the plane of the crushing plate was on average 70 mm, the distance sensor was polled 30 times per second, while the total scanning time was 2 min for a scanning working area of 600 cm^2^. The crushing plate model was visualized (Figure 8b). Although this is not the purpose or even a mandatory step of scanning, this was conducted solely for clarity of comparison with the real object. The scanning accuracy was good and new 3D model reconstruction algorithms made it possible to obtain a model with the same distances determined with an accuracy of 0.5 mm, namely 100 mm between the centers of the ribbed projections and a projection height of 30 mm. During this study, different scanning speeds were tested: from 10 cms^−1^ to 1 ms^−1^, depending on the complexity of the object geometry; however, we came to the conclusion that it is reasonable to reduce the speed of the distance sensor on the working segments of the trajectory to ensure a set of more points for statistical processing.

The MPS was carried out by moving the robot arm with the microplasmatron installed on it in accordance with the obtained 3D model of the crushing plate. The spraying distance was maintained equal to 100 mm (Table 3). As previously noted in Section 2.2, the spraying parameter was chosen based on the requirements for the structure of the coating.

As shown in Figure 8a, the substrate surface is first scanned using a distance sensor attached to the robotic arm. A robotic arm with a sensor moves in a plane along a U-shaped trajectory with a constant modulus speed. Figure 8a shows how, by measuring the Z coordinates from the surface of the plate, 3D scanning is carried out, based on the results of which a 3D model of the substrate is reconstructed (Figure 8b). In practice, the 3D model is not visualized since it is a set of point clouds, that is, three-dimensional coordinates that remain in the robot’s memory.

Then, the distance sensor on the robotic arm is replaced with a microplasmatron (Figure 8c) and the robot’s movement is generated along a 3D trajectory (that is, along three-dimensional coordinates, but with the addition of the spraying distance H to Z-coordinate). As a result, the microplasmatron moves at a given spraying distance H from the surface of the plate with strict adherence to the linear speed of movement V, while maintaining the perpendicularity of the plasma jet to the substrate. When the robot reaches the edge of the plate, it moves horizontally by a given distance—the spraying step τ—and turns, moving at the same linear speed in the opposite direction, ensuring that the spraying tracks overlap by approximately one-third. As a result, a coating of uniform thickness is sprayed (Figure 8d).

Ideally, a robot with a microplasmatron could move to the area of the worn surface itself and perform spraying, but this task has not yet been solved since additional analysis of the image of the plate surface was required to determine the boundaries of the wear area. The method implemented here did not use a video camera, did not perform image analysis, and used only distance sensor data. Therefore, in practice, the manipulator was manually brought to the boundaries of the worn area, where the coordinates were measured. Then, a code for moving the robot was generated, stopping the MPS process when the worn area was completely coated. This was a pilot experiment, during which robotic MPS of a protective coating was performed on worn areas of the surface of a crusher plate. In contrast, the movement of a robotic arm with a microplasmatron was automatically generated along the reconstructed 3D model of the product with precision maintaining a given linear speed and distance to the surface.

Production tests of the crushing plate were carried out under conditions of grinding polymetallic ores of various hardness with continuous operation of the crusher for at least 30 min with a loading from 70% to 100% of the height of the crushing chamber. The movable plate of the jaw crusher, coated after its main resource had been depleted, was additionally operated for 6 months, being subjected to the optimal load for 5–6 h per shift. Thus, the plate’s service life was extended with the same crusher performance. A production test certificate was received (Ust-Kamenogorsk, Kazakhstan), and it was concluded that the service life of the movable plate of the jaw crusher restored by robotic plasma spraying of the Cr-based protective coating was increased by 15% compared to the plates not subjected to restoration.

Thus, the practical application of the new approach to control the robot manipulator performing microplasma spraying of a protective coating on the surface of the jaw crusher plate was successful. The novelty of the proposed approach lies in the use of the technique of constructing the geodesic distance function from the starting curve, which provides two advantages in comparison with other known methods. Firstly, the proposed method is insensitive to the choice of the starting geodesic curve, whereas, in existing seed curve methods, the starting curve is often chosen as the geodesic max height. Thus, the new method offers the prospect of fully automatic trajectory generation. Secondly, this technique provides better accuracy, which enables operation with trajectories with a small step between the working segments of the trajectory, which improves compliance with technological parameters and, accordingly, the quality of the coating.

## 4. Further Perspectives

First, future research is aimed at developing a fully automatic trajectory planning procedure, when the selection of the optimal starting curve will be carried out with the development of software that selects the starting curve in the optimal way.

Second, the research will be aimed at creating a surface preliminary scanning procedure that allows working with surfaces of complex shapes and using computer vision methods when scanning surfaces.

Further development of research will also involve comparing the final product quality produced with different robot control algorithms for evaluation of methods and results using the spraying simulator.

In addition, the applicability of this method will be considered for alternative domains as well, e.g., in the finalization of 3D printed products, where some applications (in medicine particularly) require specific surface treatment [50,51]. We also plan to investigate the primary and secondary impact of the robotic approach from the sustainability point of view [52]. Many of these experiments will be conducted in the Antal Bejczy Center for Intelligent Robotics and Obuda University [53].

## 5. Conclusions

The main contribution of this research is the presentation of a novel path planning method and also a new, highly accurate technique for constructing a geodesic distance function from the starting curve. The use of the proposed method improves the quality of thermal plasma sprayed coatings.

A practical implementation of the developed path planning algorithms was carried out with the help of an intelligent system, based on an industrial robot from Kawasaki, where a microplasma spraying of a protective coating was carried out on a jaw crusher plate, which was then in operation for 6 months, crushing mineral raw materials.

The study results make it possible to increase the efficiency of the technology of robotic thermal plasma spraying of coatings, improve the performance characteristics of processed products, and “cost-effectively” produce robotic plasma spraying of coatings on parts or large-sized products, including piece products.

New developments in control algorithms are of keen interest to a wide range of researchers in the fields of robotics and automation of production areas with mechatronic systems.

## Figures and Tables

**Figure 1 sensors-25-00708-f001:**
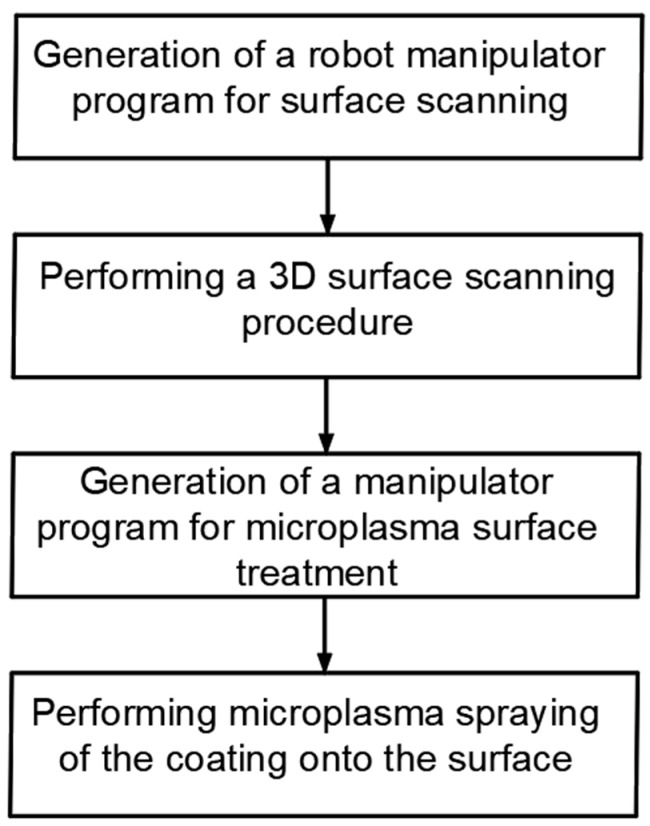
Flow chart of the process of microplasma spraying by an intelligent robotic system.

**Figure 3 sensors-25-00708-f003:**
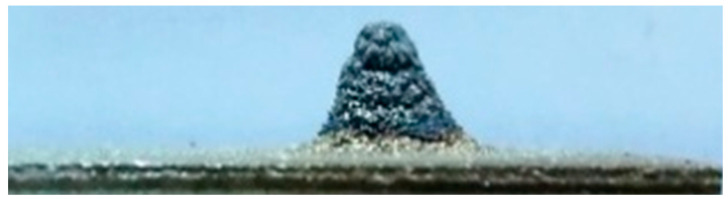
Metallization figure produced by stationary microplasma spraying of zirconium wire onto a stationary steel substrate for 10 s [35].

**Figure 4 sensors-25-00708-f004:**
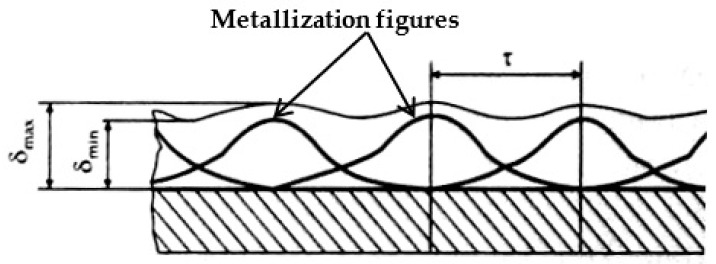
Scheme for the formation of a coating with a uniform thickness, where τ is the spraying step, δ_max_ is the largest coating thickness, and δ_min_ is the smallest coating thickness [35].

**Figure 5 sensors-25-00708-f005:**
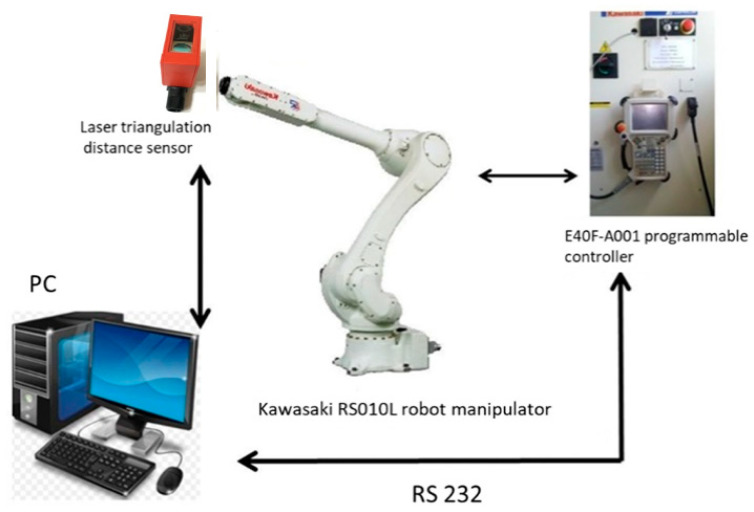
Schematic representation of robotic 3D scanning using a laser triangulation distance sensor, robotic manipulator and its programmable controller, PC, and digital interface RS 232.

**Figure 6 sensors-25-00708-f006:**
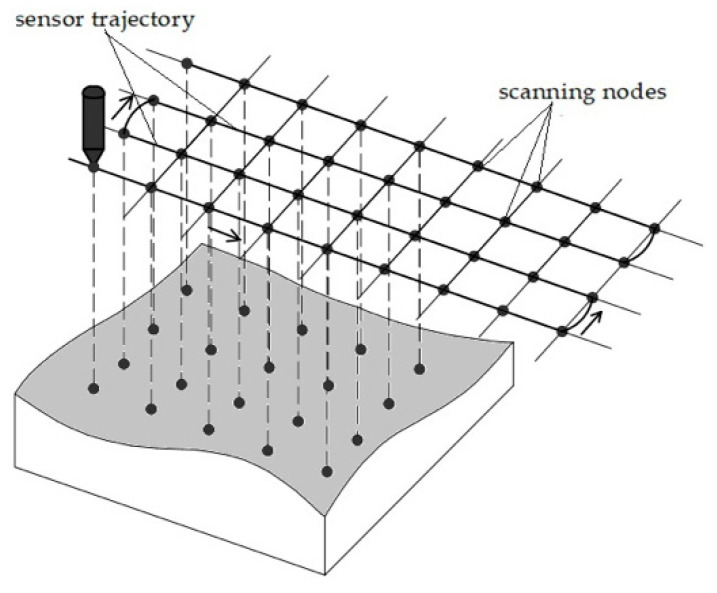
Scheme of scanning the surface of an object, indicating the trajectory of the sensor performing measurements at the scanning nodes. Arrows indicate the direction of sensor movement.

**Figure 7 sensors-25-00708-f007:**
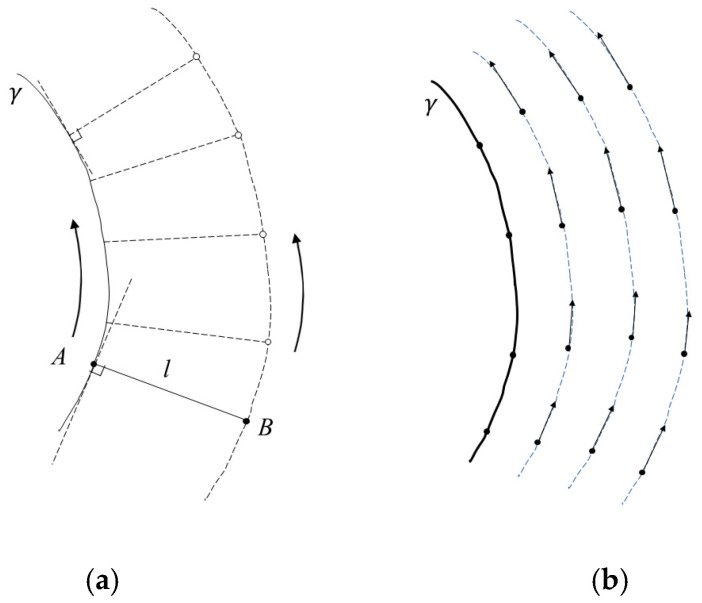
Schematic representation of (**a**)—the kinematic procedure for constructing the isolines of the vector field function $\phi_{\gamma }$, the arrows show the direction of movement of pucks A and B, *l* is the length of the string connecting pucks *A* and *B*, ***γ*** is the seed curve and the trajectory of the puck *A*; (**b**)—the corresponding tangent vector field ***U***.

**Figure 8 sensors-25-00708-f008:**
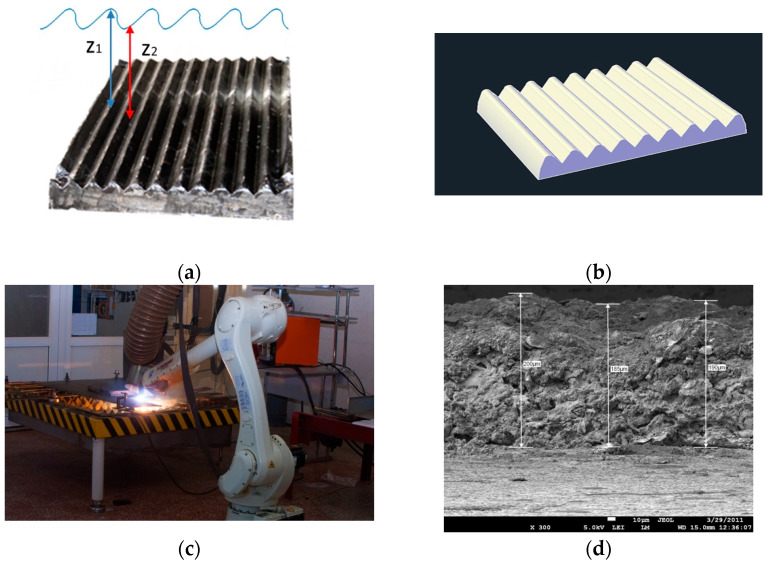
Sequence of robotic 3D scanning and subsequent MPS of Cr-based powder coating on the jaw crusher plate: (**a**)—3D scanning: distance sensor measures the distance to the surface of the plate (Zi coordinate, as Z1 and Z2); (**b**)—3D model reconstructed from scanning data to generate manipulator code; (**c**)—the end effector of the robot with the microplasmatron moves along a 3D trajectory; and (**d**)—Cr-based coating of uniform thickness.

**Table 1 sensors-25-00708-t001:** Chemical composition of steel grade 110G13L, wt.%.

Fe	C	Mn	Cr	V	Si	P	S
Base	1.19	12.1	-	-	0.39	0.021	0.015

**Table 2 sensors-25-00708-t002:** Chemical composition of AN-35 composite alloy powder, wt.%.

Co	Cr	Ni	Si	Fe	C	W
base	28–32	≤3	1.7–2.5	≤3	1.3–1.7	4–5

**Table 3 sensors-25-00708-t003:** Parameters of MPS of AN-35 powder.

MPS Parameters	Settings
Electric current, I (A)	40
Plasma gas (Ar) flow rate, Q (slpm)	6
Spraying distance, H (mm)	100
Powder feed rate, V_pow_, (g min^−1^)	2

## Data Availability

The data presented in this study are available upon request from the corresponding authors.

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
