# Peer review of "Application of Discrete Exterior Calculus Methods for the Path Planning of a Manipulator Performing Thermal Plasma Spraying of Coatings [Author-notes fn1-sensors-25-00708]"

_sensors, 2025, doi:10.3390/s25030708_

Round 1
Reviewer 1 Report
Comments and Suggestions for Authors
Sections 1 and 2 of the paper are well written and explain the process of thermal plasma spraying with a manipulator very clearly. This is also easy to understand for readers from other areas of robotics.
Section 3, on the other hand, is not very well embedded in the rest of the paper. There are often gaps in the description of the mathematical approach (for example, individual parameters are not introduced). Therefore, it is difficult for a reader who is not familiar with this mathematical approach to understand the relevance for the current problem. In particular, the final result, the calculation of a geodesic distance function, is not sufficiently transferred to the original problem. How exactly is the distance function transferred to the robot controller? With regard to the results, I would have been interested in the trajectory that was determined using the new method. The connection between the new method and the results obtained is not clear.
Comments on the Quality of English LanguageThere are several parts of the paper, where sentence structure is not sufficient. For example in line 757 (see below), the reader can not catch the summary of the subsection.
"However, the theoretical question is: what conditions are imposed on the surface the vector fields that correspond to the starting geodesic curve can be considered "approximate Killing vector fields" remains open, and the above reasoning is based on empirically obtained data in favor of the validity of the hypothesis that such vector fields are approximately Killing vector fields."
I suggest to rework the whole paper with respect to language and formatting
Author Response
We sincerely thank the reviewers for their time, effort, and attention to detail in reviewing the manuscript. Their valuable insights and constructive feedback have greatly contributed to improving the quality of the paper. We have carefully considered all the suggestions made by the referees and have incorporated the relevant changes into the revised manuscript. We believe these revisions have significantly enhanced the clarity, depth, and quality of our paper.
We have attached the revised manuscript along with a detailed response letter addressing each point raised by the referees. To improve the quality of the English, the manuscript was carefully proofread by a native speaker. All changes are highlighted in yellow. Please, see the attachments

Reviewer 2 Report
Comments and Suggestions for Authors
at first we need to consider more related and update refs in introduction section
secondly, i have many problems with your structures, necessity, and the main inovations.
finally, i want to mentioned that could you certify your results exactly?
Author Response
Answer to the reviewers
We sincerely thank the reviewers for their time, effort, and attention to detail in reviewing the manuscript. Their valuable insights and constructive feedback have greatly contributed to improving the quality of the paper. We have carefully considered all the suggestions made by the referees and have incorporated the relevant changes into the revised manuscript. We believe these revisions have significantly enhanced the clarity, depth, and quality of our paper.
We have attached the revised manuscript along with a detailed response letter addressing each point raised by the referees. To improve the quality of the English, the manuscript was carefully proofread by a native speaker. All changes are highlighted in yellow.

Reviewer 3 Report
Comments and Suggestions for Authors
This paper proposed an original geodesic distance function from the starting curve of trajectory planning for the robot manipulator, and further developed the robot system for the plasma processing of industrial products, which improves the quality of thermal plasma sprayed coatings. Besides, the proposed method was tested through robotic microplasma spraying of a protective coating on the surface of a jaw crusher plate, and successfully applied to actual industrial production. The content is quite clear and well organized, but there are some problems need to be addressed:
1. Some sentences contain grammatical mistakes or are not complete sentences, such as:
l In line 132, page 3, “pass planning” should be path planning;
l The sentence “kis a numerical co-285 efficient,” in line 285, page 7, missing spaces between symbols and text;
l In line 346, page 8, the sentence “400mm; the nonlinearity.” Is not complete;
l In line 512, page 13, “A and B” has different font;
l In page 14, there seems to be a meaningless point to the right of equation (5);
l In line 551, page 14, the format of the formula reference in this sentence “and the point 𝐵 𝑎𝑠 (7)^” seems wrong.
2. In the experimental part, the authors demonstrate the advantages of the proposed method through qualitative analysis, and the result is shown in Figure 8, but do not carry out specific quantitative analysis. It is suggested to compare with the traditional method, and give clear results of indicator(s) comparison.
Author Response

(The authors gave the same response as above.)

Round 2
Reviewer 2 Report
Comments and Suggestions for Authors
my concerns should be responded in details. i am not satisfy/
Author Response
Thank you very much for taking the time to review this manuscript. Please find the detailed responses below and the corresponding revisions/corrections highlighted in yellow (and in blue text)
Comment 1. at first we need to consider more related and update refs in introduction section
Author's response: With gratitude for this fair comment, we have considered more different approaches to solve the current problem in the introduction section and increased the number of relevant references. We have made a corresponding update, adding 6 new references related to the problem being solved as follows:
P.3 Line 124-135: Nowadays, the literature describes methods for generating a manipulator trajectory for spray painting optimized for various types of 3D models. For example, for the common STL 3D model format, Wu et al. developed a boundary fitting approach [21]. The raw 3D scanning data is a point cloud. The methodology of manipulator path generation based on point cloud segmentation is presented in the article by Hua et al. [22]. A very promising approach using a machine vision system for online manipulator path generation when scanning 3D objects was presented by Maс et al. [23]. It should be noted that 3D surface metrology, sensing, point cloud data generation/ analysis, 3D surface generation, and path planning and trajectory generation are effectively similar for painting [8-12] [14-16] [21], fiber placement [24] [25], laser surfacing, ploshining, and plasma spraying [18], therefore, special attention will be paid to the consideration of these approaches, in particular the approach used for trajectory generation for robotic fiber placement [25].
[21] Wu, H.; Tang, Q. Robotic spray painting path planning for complex surface: boundary fitting approach. Robotica 2023, 41, 6, 1794-1811.
[22] Hua, R. X.; Ma, H-X.; Zou, W.; Zhang, W.; Wang, Zh. A Spraying Path Planning Algorithm Based on Point Cloud Segmentation and Trajectory Sequence Optimization. International Journal of Control, Automation and Systems 2024, 22, 2, 615–630.
[23] Mac, T. T.; Trinh, D. N.; Nguyen, V. T. T.; Nguyen, T-H., Kovács, L.; Nguyen T-D. The Development of Robotic Manipulator for Automated Test Tube. Acta Polytechnica Hungarica 2024, 21, 9, 89-108.
[24] Aized, T.; Shirinzadeh, B. Robotic fiber placement process analysis and optimization using response surface method. Int. J. Adv. Manuf. Technol. 2011, 55, 393-404.
[25] Shirinzadeh, B.; Cassidy, G.; Oetomo, D.; Alici, G.; Ang Jr, M. H. Trajectory generation for open-contoured structures in robotic fibre placement. Robotics and Computer-Integrated Manufacturing 2007, 23 (4), 380-394.
Comment 2 secondly, i have many problems with your structures, necessity, and the main inovations.
Author's response: We appreciate the reviewer's comments, and we have made changes to improve the structure of the manuscript and to highlight two major innovations in this paper:
1) Calculation of the geodesic distance function from the seed curve and further construction of offset curves as isolines of this function, which gives a significant advantage in the accuracy of trajectory construction
2) An original method for calculating the geodesic distance from a given curve on the surface is proposed, characterized by high speed with sufficient accuracy
Specifically, we added the following clarification to section 3.2 Manipulator path planning at lines 473-481: Of course, the general approach to manipulator trajectory planning, which involves constructing the manipulator path on a surface, is quite common for robotic spraying applications in painting, varnishing, etc., as mentioned in the introduction section. However, the problem of constructing the manipulator path on the surface is closely related to the problems arising in the tasks of robotic fiber placement, more precisely, there are a number of trajectory planning tasks for resin transfer molding (RTM), automated tape placement (ATP) and robotic fiber placement (RFP) [25]. The interrelations between these trajectory planning methods and the proposed method will be discussed in more detail in the next section.
We have made changes to the structure of the manuscript by adding the following subsection titles:
3.2.1 Offset Curve planner approach and geodesic distance function
3.2.2 Isolines of geodesic distance function as integral lines of vector fields
We have included the following explanation in subsection 2.3.1, lines 492-503: Similarly, the Surface Curve Offset (SCO) algorithm for RFP presented by Shirinzadeh et al. in [25] is based on an approach that can be summarized as follows: “For a given RFP path, the next path must be offset along the surface by a distance of one tow-width in the perpendicular direction from the given path.” The distinctive feature of the SCO algorithm is that the offset of a given point of the curve along the surface (in a given direction) is determined by the intersection of the so-called control plane with the surface, and the New-ton-Raphson root-finding method is used to solve the surface-plane intersection equation (strictly speaking, the initial path is also formulated from the surface-plane intersection). Thus, the SCO algorithm proposed by Shirinzade et al. in the article [25] is applicable to the problem of constructing trajectory traces and may achieve better results in terms of accuracy than previously proposed curve shift methods [13].
Comment 3 finally, i want to mentioned that could you certify your results exactly?
Author's response: To address this concern of the reviewer, we provided quantitative data on assessing the benefits of the proposed method as follows:
- 25 Line 972-999: To test the software implementation of the described method for constructing the geodesic distance function from a curve, we used both synthetic data (triangular meshes calculated for analytically defined surfaces) and data obtained from scanning. It should be noted that most of the available 3D mesh libraries (such as MeshLib) do not contain functions that calculate the geodesic distance from a given curve on the surface. Typically, these libraries provide functions that calculate the geodesic distance between two given points on the surface or, at best, numerically calculate the geodesic distance function from a given grid vertex. Thus, using independently developed libraries to quantitatively assess the accuracy of calculating the geodesic distance function using the algorithm described above is difficult. For such a comparative assessment, we used our own software implementation of the geodesic heat method. We extensively used the library of computational geometry methods to create a software implementation of the geodesic heat method [47]. As shown by the results of numerical experiments, the relative error in calculating the geodesic distance from a point on the surface to a given curve using the algorithm described above in most cases does not exceed 0.7% and is practically independent of the position of the point relative to the curve. Considering that we compare two approximate methods, our algorithm and a geodesic heat method, we believe that the accuracy of the proposed method is more than sufficient for trajectory planning tasks. Note that from a practical point of view, for complex surfaces, a good choice of the seed curve has a much greater impact on the final result of the surface processing procedure than the accuracy of calculating the geodesic distance function.
With these modifications and additions, we truly believe that the manuscript has been largely improved, and hoping to find acknowledgment from the reviewers as well.